# Recreation Potential Assessment at Tamarix Forest Reserves: A Method Based on Multicriteria Evaluation Approach and Landscape Metrics

Mahmoud Bayat [1,*], Pete Bettinger [2], Sahar Heidari Masteali [3,*], Seyedeh Kosar Hamidi [4], Hafiz Umair Masood Awan [5,6] and Azam Abolhasani [7]

1  Research Institute of Forests and Rangelands, Agricultural Research, Education and Extension Organization (AREEO), Tehran 14968-13111, Iran
2  Warnell School of Forestry and Natural Resources, University of Georgia, Athens, GA 30602, USA
3  Department of Environment, Faculty of Natural Resources, University of Tehran, Tehran 14179-35840, Iran
4  Department of Forestry, Faculty of Natural Resources, Sari Agriculture Sciences and Natural Resource University, Sari 48181-68984, Iran
5  Helclean Consulting Services, Asiakkaankatu 6B 29, 00930 Helsinki, Finland
6  Department of Forest Sciences, Faculty of Agriculture and Forestry, P.O. Box 27, Latokartanonkaari 7, University of Helsinki, 00014 Helsinki, Finland
7  Desert Control and Management, Department of Reclamation of Arid and Mountainous Regions, Faculty of Natural Resources, University of Tehran, Tehran 14179-35840, Iran
*  Correspondence: mbayat@rifr-ac.ir (M.B.); saharheidari@ut.ac.ir (S.H.M.)

**Abstract:** The purpose of this study was to develop new methods to describe outdoor recreation potential based on landscape indicators and systemic multicriteria evolution in the Tamarix forest reserves of Varamin city, a part of Iranian–Turanian forests of the Tehran province in Iran. First, in conducting a multicriteria evaluation, ecological factors that included slope, aspect, elevation, vegetation density, precipitation, temperature, and soil texture were mapped, classified, and coded according to the degree of desirability for outdoor recreation. All these maps were then intersected and the final map of recreational potential for three regions of the forest reserves was prepared. Results showed that the Shokrabad region had more recreation potential than the other two regions (Fakhrabad and Dolatabad) in terms of the sum of ecological factors potentially affecting tourism potential. Second, in conducting a landscape-based method, six of the most important indicators of the landscape that are effective in outdoor recreational potential were developed for each region. The combination of these landscape features determined the value of a place for recreational activities from a landscape perspective. The results showed that a large part of the Shokrabad region and a smaller number of places in the Fakhrabad and Dolatabad regions have high outdoor recreational potential. The area suitable for recreation in the output of the multicriteria evaluation method turned out to be greater than the area suggested by the landscape method, as more factors were examined in the multicriteria evaluation method. Of the set investigated, the topography and soil factors played an important role in the evaluation.

**Keywords:** potential assessment; Iranian–Turanian forests; Varamin city; outdoor recreation





## 1. Introduction

Activities conducted for leisure or education that rely on natural environments are referred to as outdoor recreation activities. These activities are voluntary and are performed during one's free time, outside of normal work [1]. Outdoor recreation is known as one of the most important cultural activities, and these involve enjoying nature and visiting historical and cultural places. These activities are carried out in environments such as forests, fields, coastal areas, and parks [2–4]. In the past, the suitability of an area for extensive outdoor recreational activities was often evaluated with methods that required photo

collection, surveys, and interviews, such as checklists or traditional methods involving the manual intersection of ecological maps [5]. Despite the fact that these methods are useful for evaluating outdoor recreational activities, there is currently a demand for quantitative and computational methods based on geographical and quantitative data [6]. Classic methods involving quantitative databases can be used to evaluate the process of landscape change and potential future conditions, and by using these databases, it may be possible to judge the outcomes of potential changes and the success or failure of management decisions. These methods are sometimes more accurate and precise than contemporary computer-based methods and may produce results closer to reality. However, both classic and new methods may be complementary and both may be needed to evaluate the recreation potential of a place [7,8].

Advanced quantitative methods can guide in identifying areas that have potential to be used as a recreational land. The development of well-positioned recreational areas could reduce excessive pressure on the valuable surrounding forests and prevent soil erosion and desertification of these areas. In addition, valuable plant and animal species of these areas can be protected with the prosperity of tourism in these areas, resulting also in the incremental improvement in the income of indigenous people [9]. The financial benefits to a local community can potentially reduce excessive pressure that is placed on the pastures and forests of that area. Therefore, we need new methods that can simultaneously identify and integrate a range of environmental factors to help evaluate the suitability of an area for outdoor recreation purposes.

Outdoor recreation is also referred to nature-based recreation or soft ecotourism that includes recreation such as fishing, hiking, cycling, horse riding, and wildlife watching tours [10,11]. Therefore, it is clear that for such activities, the quality of the landscape can be very important. Environmental conditions such as weather, ambient temperature, vegetation status, and hydrography are important in this regard [10,12–15]. Land use also plays an important role; forest landscapes generally have a positive effect on outdoor recreation potential, whereas urban and agricultural landscapes generally may have a negative effect on outdoor recreation potential. Therefore, in order to evaluate the suitability of areas for outdoor recreation, there is a need for methods that consider a variety of landscape characteristics.

Several studies have been conducted to assess outdoor recreation potential around the world. Fangyong [16], for example, evaluated and compared the ecological potential of recreation opportunities in 12 provinces in western China. Choudhury et al. [17] used a systematic approach to locate the higher quality places for outdoor recreation activities in India. These approaches involve a geographical analysis for selecting the most suitable places, and often this involves the use of a Geographic Information System (GIS) and methods for overlapping ecological factors. Weyland and Laterra [18] studied recreational potential assessment in Argentina using a method based on the ecosystem services approach and landscape metrics. The ecosystem approach makes explicit the link between the status of natural resource systems and ecosystem services that support human wellbeing [19], and landscape metrics are measurable units of landscape composition and act as a surrogate for change, thus allowing for the description and quantification of spatial patterns and ecological processes over time and space [20,21]. It seeks to maintain the integrity and functioning of ecosystems as a whole to avoid rapid undesirable ecological change. Dağıstanlı et al. [22] assessed the suitability of land in southern Turkey for outdoor recreation using a multicriteria model, using a linear combination technique and a hierarchical analysis in association with GIS to rank the suitability of a mosaic of contiguous, seminatural sites for outdoor recreation. Zhang et al. [23] reported on the development of computer and distance-perception techniques for the management, planning, and conservation of recreation in urban green spaces.

A landscape, from a recreation perspective, is a physical setting viewed from a distance [24]. People value the quality of a landscape using multiple metrics, from its scenic beauty to its potential for human interaction when visited. Across North America, the most

widely used recreation model is the recreation opportunity spectrum, which was designed to help people understand the location and distribution of diverse recreational settings across a landscape [25]. Modifications extend the model to water resources and assessments of air, noise, and other aspects of landscape settings [26]. Although the recreation opportunity spectrum was designed for public lands of the Western United States, other models have since been designed for landscapes more heavily fragmented by private land ownerships (e.g., [27]). In general, these models attempt to place a value on the importance of a place to the human visitor. Forest parks are a forest ecosystem of natural, seminatural, or artificial origin and are used for various purposes such as recreation, nature conservation, and, in exceptional circumstances, wood production. These are covered with trees, although other elements such as water, beaten paths, and open spaces of various sizes are often found in them. Therefore, a manmade forest can be considered as a compact and rich mass of trees and shrubs, which consists of a combination of landscapes such as open spaces, open trees, closed and semiopen trees, and entrances. The open spaces and water bodies within these forests have a special role and could contribute to the nature tourism potential of a country. Nature tourism refers to a type of tourism in which nature and the environment takes precedence over everything else of interest to the recreationist. Iranian–Turanian forests have a special place in the economic and ecological development of nature tourism in the country and further influence the country's soil and water resources. Unfortunately, at present, the only use of these important resources is still the exploitation of wood and shrubs by local communities. Iran has a potential to utilize these forests similar to many other countries in the world to develop a nature tourism industry. However, according to some estimates, the recreation and nature tourism industry could grow by about 10.5% percent by 2030, and the number of nature lovers could further grow by 50%. Evaluating the suitability of an area for recreational activities requires a large amount of spatial data that includes various ecological, economic, and social factors (the accessibility of the area, the condition, type, and number of roads, and the general condition of the transportation system) [28,29]. A systems analysis method with multicriteria evaluation and with the help of Geographic Information Systems is a useful tool for this type of environmental evaluation that is used in this research [30,31]. Additionally, according to various landscape criteria and natural endowments, protection of recreational land must be assessed. This basic approach to planning without the consumption of natural resources ensures the rational use of natural assets [23,32–34].

Therefore, in this research, we aim to develop new methods to describe outdoor recreation potential based on landscape indicators and systemic multicriteria evolution in the Gaz forest of Varamin city as a part of the Iranian–Turanian forests of the Tehran province in Iran. We define landscape recreation potential based on Chan et al. [35] as the provision of outdoor recreation opportunities by natural landscapes.

In this research, we address these questions:

1. Among the studied areas, which parts have the potential for outdoor recreation?
2. Among the ecological factors examined in this research, which factor plays a more important role in evaluating the suitability of the area for outdoor recreation?
3. What are the differences in the final output between the two methods: the multicriteria assessment and the landscape approach?
4. Can the combination of two methods (the multicriteria evaluation and the landscape approach) be used as a suitable single method to evaluate the suitability of the area for outdoor recreation?

## 2. Materials and Methods

### 2.1. Study Area

The 1208 ha study area includes the Javadabad part of the Behnam district, including three forest reserves (Shokrabad, Fakhrabad, and Dolatabad), which are located about 35 km south of Varamin city on the Tehran–Mashhad railway (Figure 1). The climate of the region is a temperate warm desert climate, evaluated using the De Martonne method.

The region does not have a permanent river, yet seasonal water flows occur in the Band Ali Khan and Gol-e-Ab regions. The average annual temperature in the region is 21.3 °C and the average annual precipitation is 80 mm. The dry period in this region lasts 10 months, and only the last month of autumn and the first month of winter are generally wet periods for this region. Further, there is a possibility of frost in 5 months of the year. Wind data from the Varamin meteorological station indicate that the prevailing wind direction is southwest. This study area is low and relatively flat, with a maximum elevation of 855 m above sea level in the northern elevations and a minimum of 781 m (Figure 1) in the lowest point. The average elevation above sea level is 818 m.

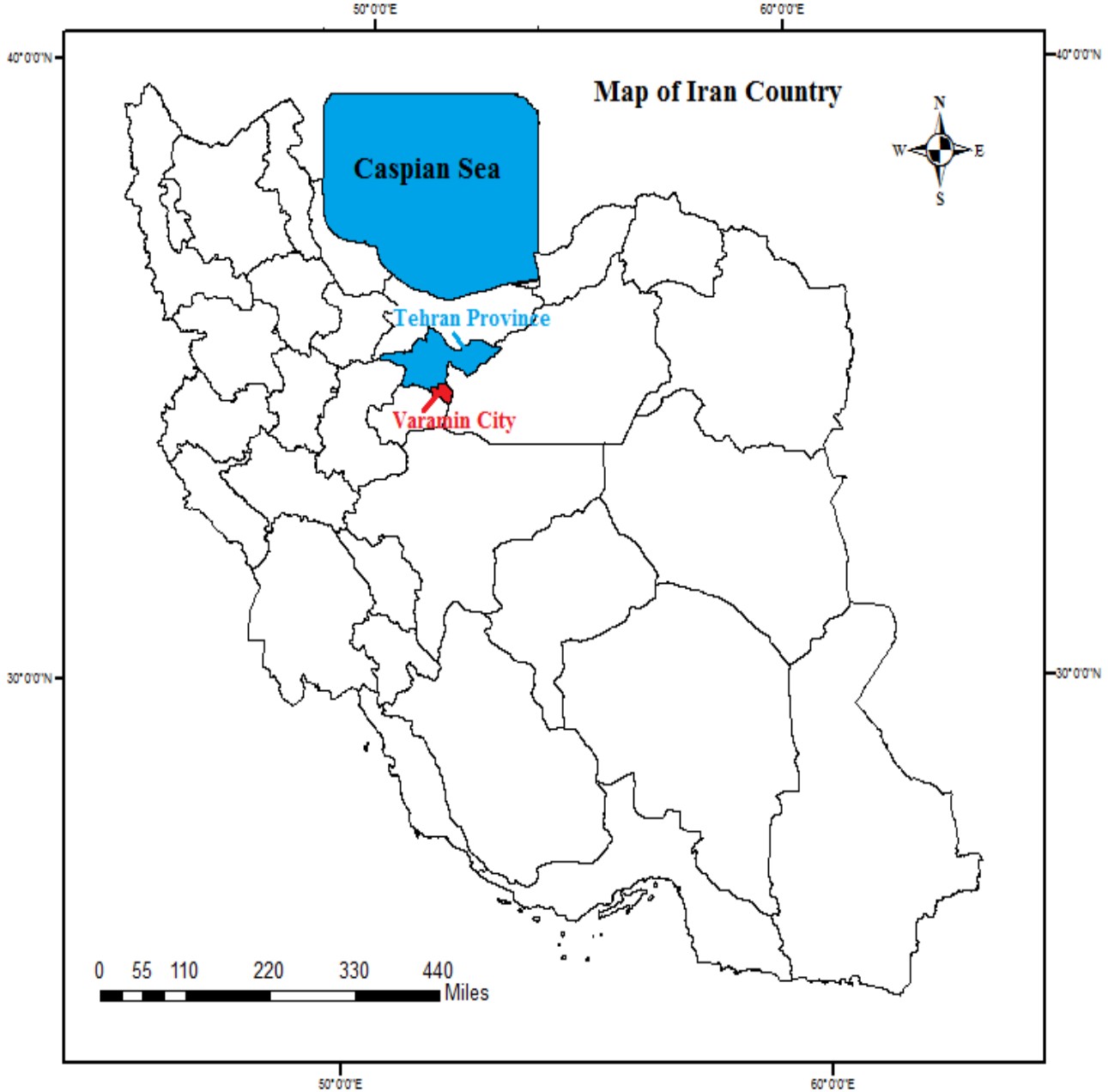

**Figure 1.** Location of the study area: location of Varamin city in the country (The blue color is Tehran province and the red color is Varamin city).

In this area, five plant types were identified as follows:
Type 1: *Artemisia sieberi + Salsola tomentosa + Tamarix* spp.

In this type (abbreviated as: Ar. si. + Sa. to. + Ta. Spp.), plain heather, salt grass, and sedge were identified as dominant species.

Type 2: *Haloxylon persicum + Annual plants + Artemisia sieberi*

In this type (abbreviation: Ha. pe. + Annual plants + Ar. Si), Zardtagh, all kinds of one-year species and plains sedge were identified as dominant species.

Type 3: *Artemisia sieberi + Alhagi maurorum*

In this abbreviated type (Ar. si. + Al. ma), the Iranian plain and Kharshter were identified as dominant species.

Type 4: *Artemisia sieberi + Salsola tomentosa + Calligonum comosum + Seidlitzia rosmarinus*

In this type (abbreviated as: Ar. si. + Sa. to. + Ca. co. + Se. ro), plains grass, salt grass, skanbil, and ashnan were identified as dominant species.

Type 5: *Salsola* spp. *+ Aeluropus littoralis*

In this abbreviated type (Sa. spp. + Ae. Li), types of grass were identified as dominant species.

The main steps of the methodology proposed in this study are shown in Figure 2 and are summarized as follows.

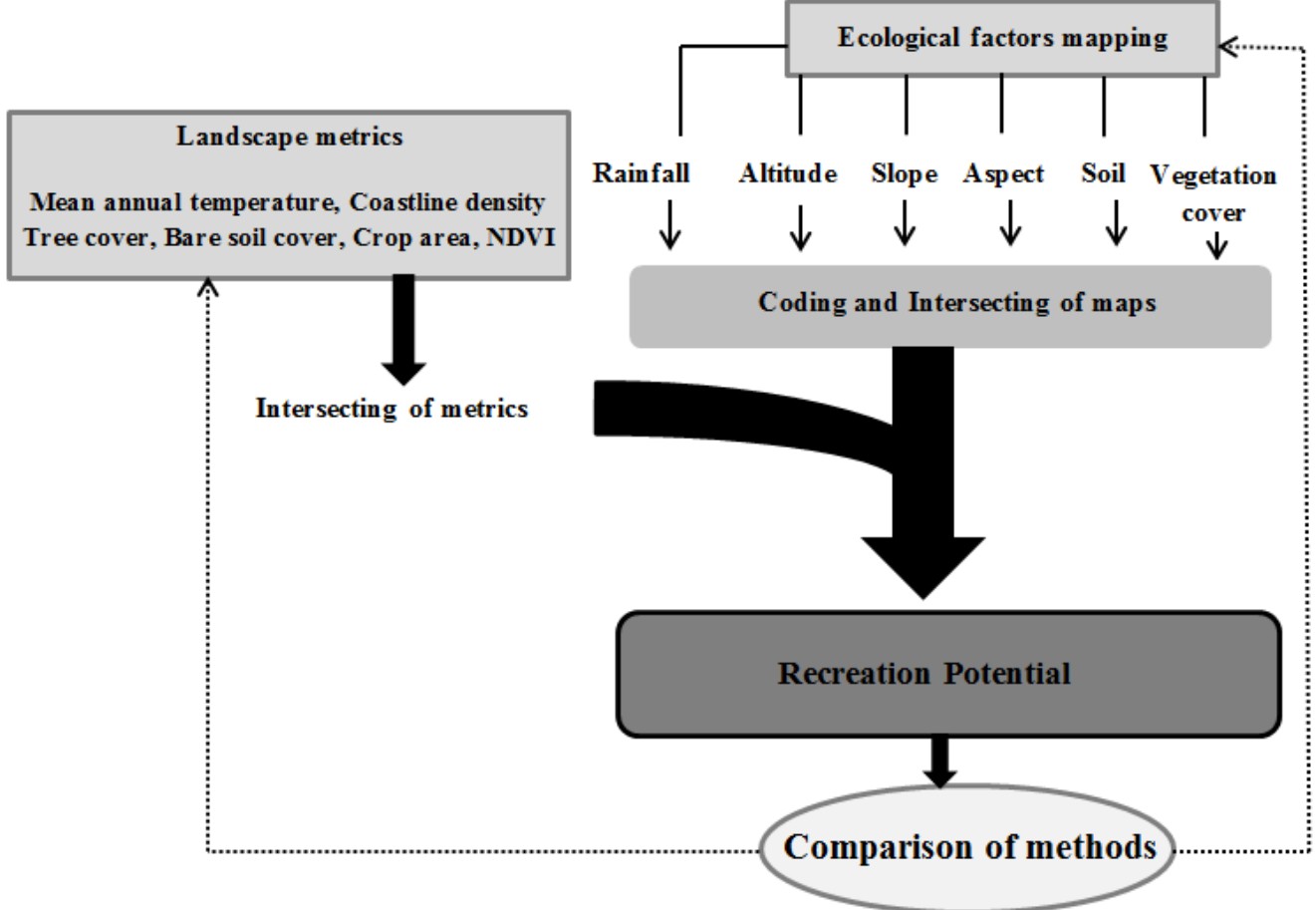

**Figure 2.** Conceptual framework of research method.

Determining ecological factors affecting tourism development (independent variables):

In the process of analyzing the environmental potential of the land for tourism development, the selection of effective environmental criteria is of particular importance. The most important effective criteria in this section are elevation, slope, direction, soil texture, vegetation density, soil fertility, and climatic factors (such as temperature, precipitation, wind speed, minimum temperature in the coldest season, and maximum temperature in the warmest season) (Table 1).

**Table 1.** Ecological factors used to evaluate outdoor recreation potential.

| Variable | Source | Expected Effect on Recreation Potential |
|---|---|---|
| Elevation | The DEM map (year of 2020) was extracted from the https://urs.earthdata.nasa.gov/ site (accessed on 8 February 2020) | Elevation is an important feature in environmental studies because it has a direct effect on other ecological factors such as precipitation and temperature [36]. |
| Slope | DEM, along with ArcGIS 10.3 software, to develop a slope map. | The morphology and slope of the basin is one of the main parameters in environmental studies that, in addition to a direct effect on the intensity of water flow, also affects other physical characteristics of the basin [37]. |
| Aspect | DEM and the Aspect tool was used in the ArcMap environment, and after creating the aspect map and necessary corrections, such as removing very small directions and merging them with larger directions. | The aspect of the domain and its changes play an important role in changing its environment. The most important effect of the aspect of the slope manifests itself in the form of differences in weather conditions [38]. |
| Temperature and rainfall | The temperature and rainfall map was taken from the regional meteorological department; it included the isolines with ArcMap. | Climate is one of the most important geophysical factors that play an important role in the establishment or nonestablishment of human settlements. Therefore, the climate is an important criterion in the establishment of tourist centers [39]. |
| Vegetation density | Field visiting, report, and mapping by ArcMap. | The higher the vegetation density, the higher the tourism potential for this purpose [40,41]. |
| Soil | Field work, sampling with plots, lab analysis, and mapping by ArcMap. | Soil erosion is one of the most fundamental environmental, agricultural, and food production problems in the world that has devastating effects on all natural and fabricated ecosystems. Soil loss is, therefore, recognized as a serious environmental problem [42,43]. |

Mapping, classification, and coding of ecological factors:

After determining the ecological factors, the next step includes mapping all the mentioned ecological factors and extracting spatial data.

1. Elevation:

To investigate this factor in the study area, a digital elevation model map of the area was used and the points were classified and coded based on the division described in Table 1.

The DEM map (year of 2020) was extracted from the https://urs.earthdata.nasa.gov/ site with a resolution of 12.5 m and entered into the ArcMap environment. This map was the base map of slope and aspect maps.

2. Slope:

To calculate the slope and prepare the slope map of the regions, the DEM was used, along with ArcGIS 10.3 software, to develop a slope map. The slope map was prepared automatically and in raster format based on the data of DEM and using Slope tool in ArcMap. Then, using Reclassify tools, this map was divided into the desired classes specified in Table 1.

3. Aspect:

In order to prepare this map, the DEM was used, along with ArcGIS 10.3 software, to develop an aspect map (Table 2). The aspect map was made based on the base map of DEM. To make this map, the Aspect tool was used in the ArcMap environment, and after creating the aspect map and necessary corrections, such as removing very small directions and merging them with larger directions, the final map included four main directions, plus a plane or without-direction class (slopes of less than 5% are known as no direction) was created. These maps entered the next stage for analysis and coding.

**Table 2.** Classification and coding of factors of elevation, slope, and geographical aspect.

| Desirability Code | Geographical Aspect | Desirability Code | Slope (%) | Desirability Code | Elevation (m) |
|---|---|---|---|---|---|
| 1 | North | 0–2 | 1 | 1 | >1000 |
| 1 | Eastern | 2–5 | 2 | | |
| 2 | South | | | | |
| 2 | Western | | | | |

4.    Temperature and rainfall:

A climate map was prepared from the data of Varamin meteorological stations. Temperature and precipitation were extracted and isolated in three study areas. The temperature and precipitation map that was taken from the regional meteorological department included the isolines of both temperature and precipitation in the studied area, which entered the ArcMap environment [44]. In terms of temperature, almost the entire study area was in the temperature range of 21–25 degrees. In terms of precipitation, the study area had three layers of precipitation. A part of the area between the precipitation isolines was less than 40 mm. A part between the isolines of rainfall was 60–40 mm and a part between the isolines was more than 60 mm. Then, using these two maps, the coding of climatic factors in the study area was performed according to Table 3.

**Table 3.** Classification and coding of precipitation and temperature factors.

| Desirability Code | Temperature (°C) | Desirability Code | Rainfall (mm) |
|---|---|---|---|
| 1 | 21–25 | 3 | >40 |
| | | 2 | 40–60 |
| | | 1 | More than 60 |

5.    Vegetation density:

Vegetation density in each part of the region was measured and recorded by field visits, and then the vegetation density code was assigned to each class. To prepare a vegetation density map using plots, and measuring the small and large diameter of the vegetation tree in each plot, the vegetation density area of each plot was determined and this was performed for all parts of the studied area. These data were then entered into an ArcMap environment and, based on the plot coordinates and the calculated canopy area, a map of the relative vegetation density of the area was prepared and classified and coded, according to Table 4, as in the other maps.

**Table 4.** Classification and coding of vegetation density factor.

| Desirability Code | Vegetation Density (%) |
|---|---|
| 1 | More than 40 |
| 2 | 20–40 |
| 3 | 0–20 |

6.    Soil:

In this study, a sufficient number of soil samples were taken from the three study areas up to a depth of 100 cm and the coordinates of each soil sample were collected with GPS with appropriate accuracy as the area lacked dense canopy [45–47]. In this regard, according to laboratory analyses, soil samples were extracted from the soil texture map and we recorded organic carbon percentage and fertility, which received loamy texture code 1 and loamy texture code 2; silty loamy clay texture was found to be undesirable.

After the laboratory analysis data were taken for each soil sample, these data were entered into the ArcMap environment along with the coordinates of each point where the

soil samples were taken. Then, the point layer of the soil characteristics of the area was prepared, the points with similar characteristics formed a polygon, and the final soil layer of the area containing several layers or soil polygons with different soil characteristics was created. These classes were coded based on the appropriateness or lack of appropriateness of the soil characteristics they had (Table 5).

**Table 5.** Landscape metrics (variables) used as indicators of outdoor recreation potential [18].

| Variable | Description | Expected Effect on Recreation Potential | Source |
|---|---|---|---|
| Mean annual temperature | The average annual temperature of each region was calculated and determined using isothermal lines. | Temperature plays an important role in outdoor recreation, as extreme temperatures restrict activities such as camping. [48,49]. | Information of meteorological stations in the region (map of isothermal lines of the region). |
| Coastline density | River, streams, lakes (km coastline/km$^2$). | Rivers allow activities such as swimming and fishing [14]. | Hydrographic layer of study area. |
| Tree cover | Percent tree cover. | Vegetation percentage is one of the most important factors in outdoor recreation because this factor determines the shade percentage [50,51]. | Satellite images of the study area. |
| Bare soil cover | Percent bare soil. | If the proportion of bare soil in the area is high, it will cause problems for activities such as camping due to the lack of shade and harsh weather conditions [28]. | Satellite images of the study area. |
| Crop area | Percent herbaceous and shrub crops and forestations. | Human activities such as agriculture diminish scenic value and recreation activities [52]. | Satellite images of the study area. |
| NDVI | Normalized Difference Vegetation Index as an indicator of lush vegetation. | Landscapes with lush vegetation are preferred by recreationists [52]. | Satellite images of the study area. |

Determining and classifying the ecological potential of tourism:

After mapping all the desired ecological factors and extracting the utility codes and stacking all the maps together, finally, a general index for each of the integrated environmental units in each of the three areas was obtained, for which code 1 was considered as having suitability, code 2 was evaluated as having recreational potential, and code 3 as without potential.

### 2.2. Landscape Approach

Environmental factors such as the physical conditions of the land, weather and especially temperature, landform, the percent of tree cover, and hydrography are the identified factors that determine the potential of an area for outdoor recreation [10,14,15]. The combination of these features determines the value of a place for recreational activities from the landscape perspective. Therefore, in the present study, an attempt was made to review and determine a set of the most important environmental factors by reviewing the sources (Table 4). Therefore, each region is a landscape unit on which we estimated six landscape metrics (Table 4).

### 3. Results

All ecological factors were mapped, classified, and coded, which can be seen in Figures 3–5. Figures 3–5 show the set of maps of ecological factors created and classified in Shokrabad, Fakhrabad, and Dolatabad regions, respectively. Figure 6 shows the final map resulting from combining the mapped ecological criteria, which is the final map of outdoor recreation potential. Based on the classification, the different parts of each region are divided into three classes: no power, class 2 power (middle capacity), and class 1 power (good capacity). Based on this method, only parts of the south and southeast of Shokrabad and west of Dolatabad were recognized as having a good capacity for recreation.

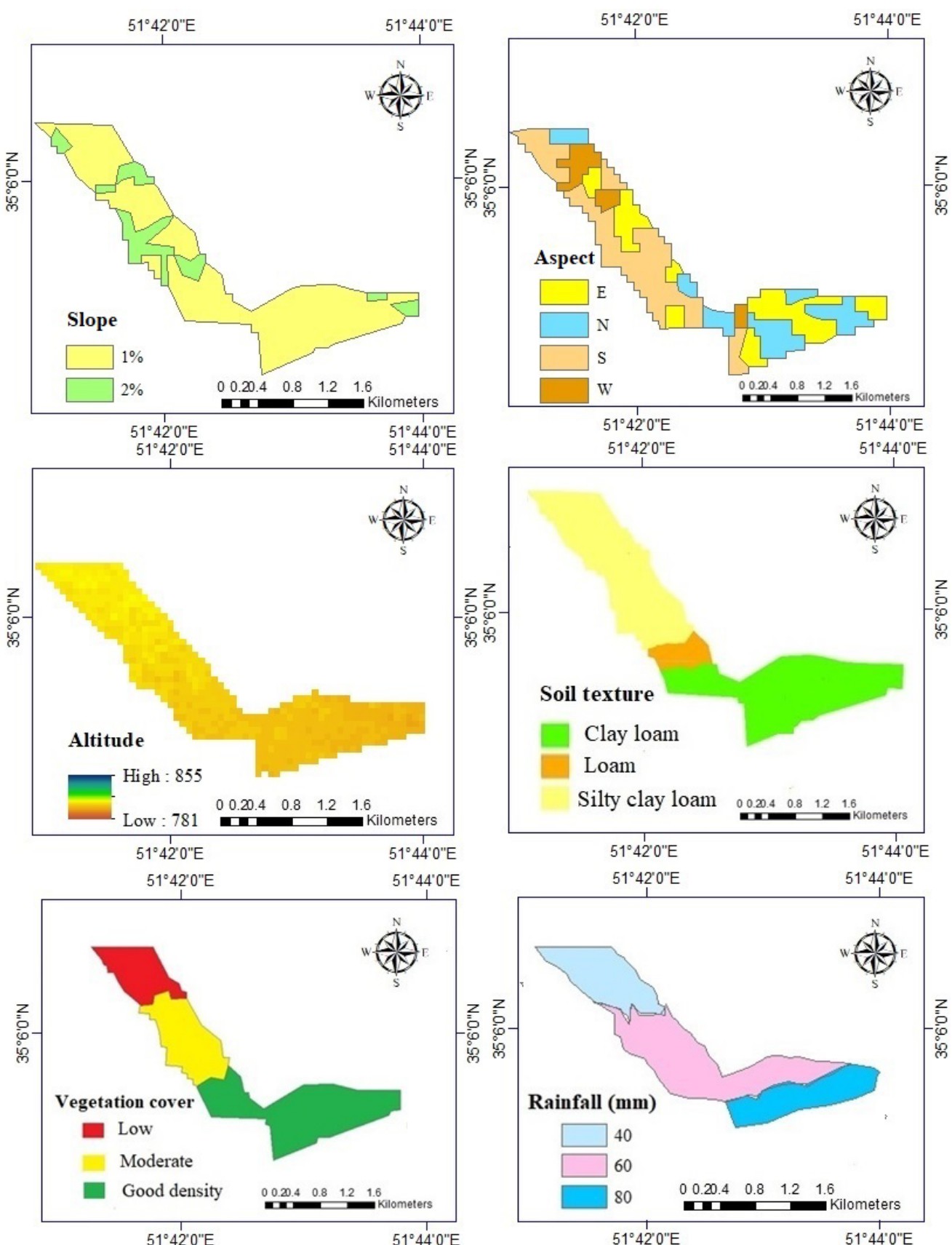

**Figure 3.** Classification map of ecological factors in Shokrabad region.

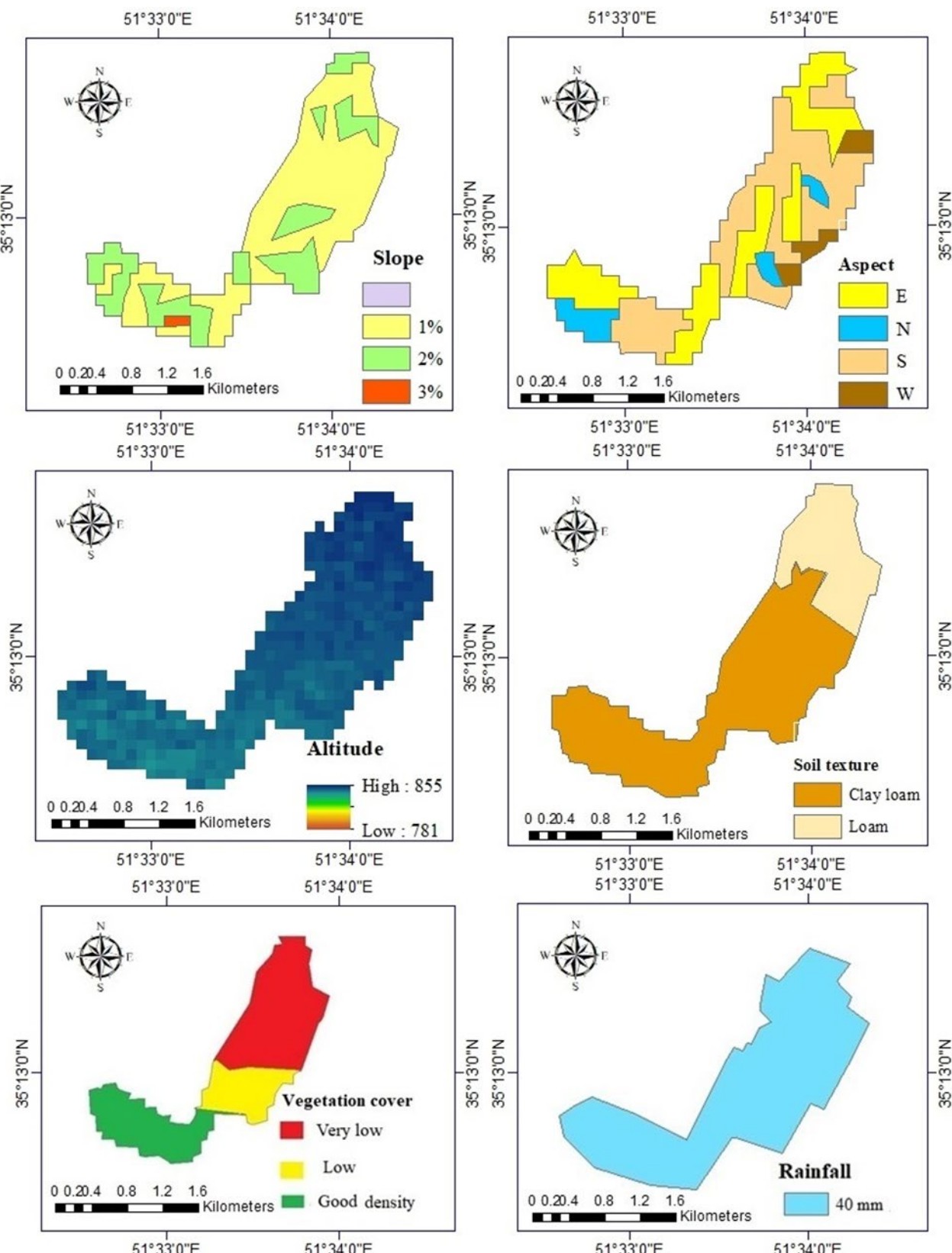

**Figure 4.** Classification map of ecological factors in Fakhrabad region.

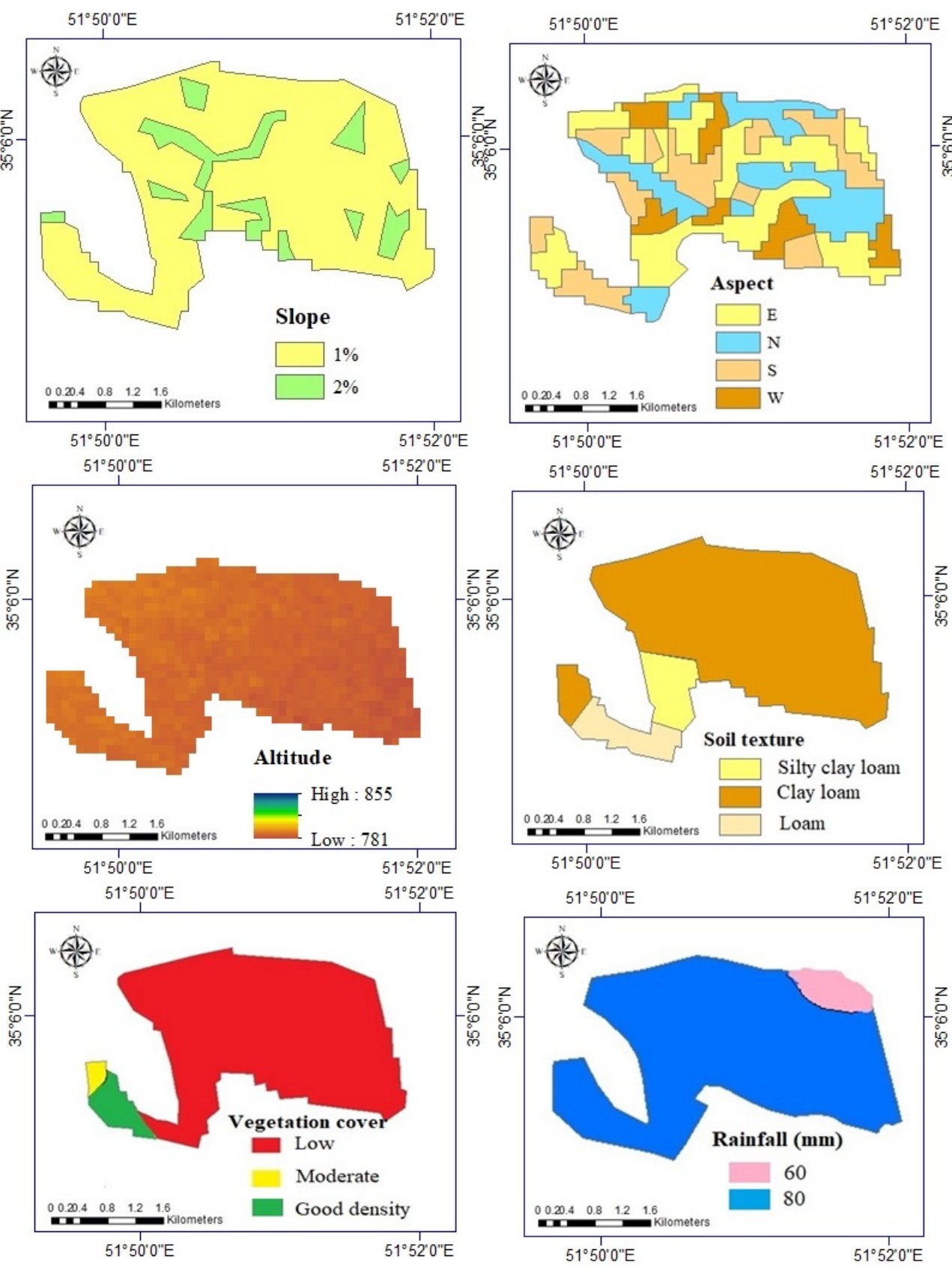

**Figure 5.** Classification map of ecological factors in Dolatabad region.

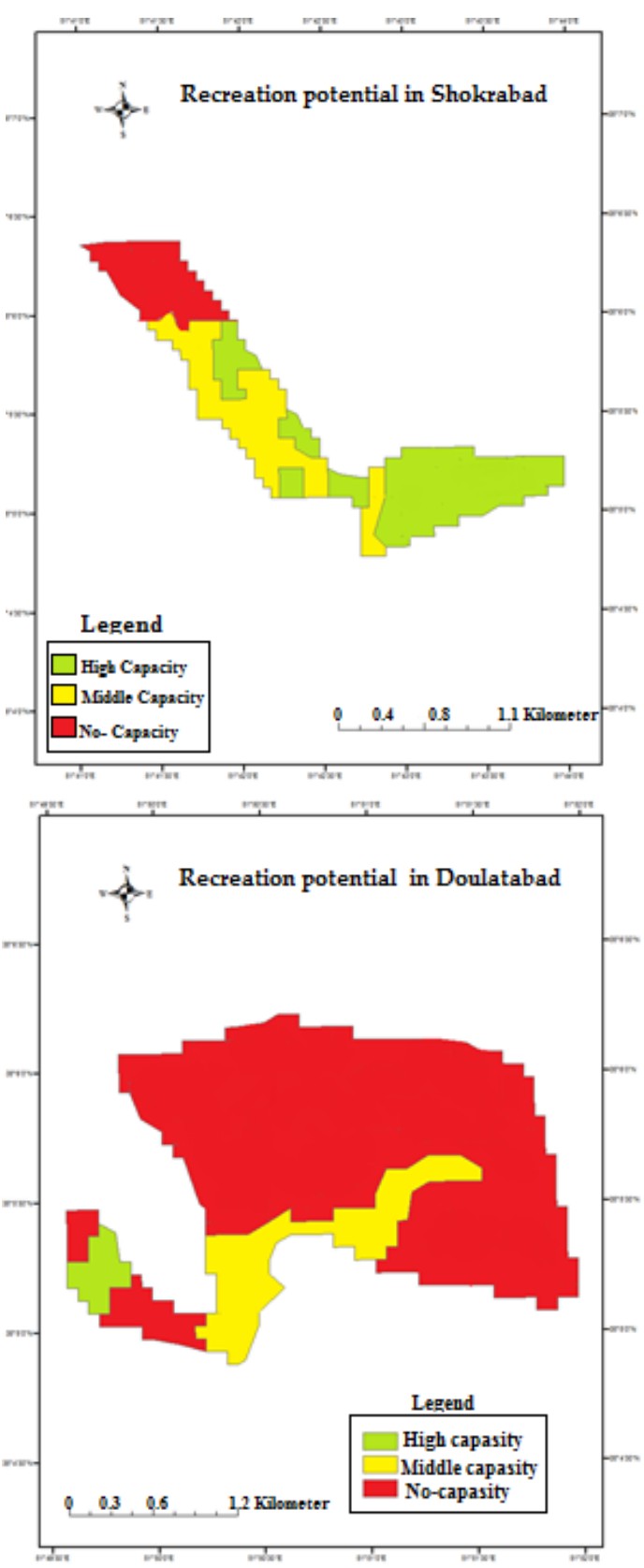

**Figure 6.** *Cont.*

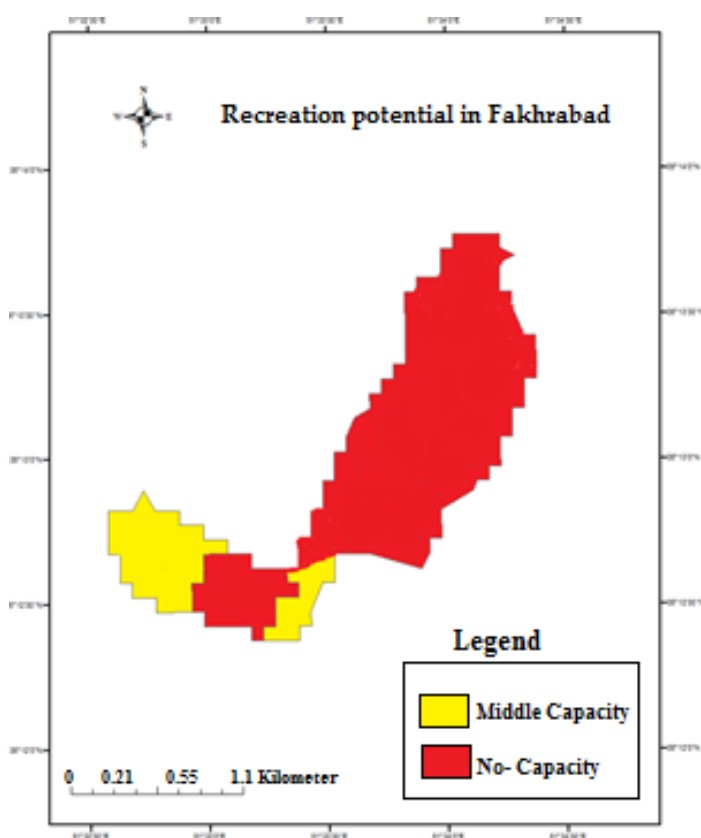

**Figure 6.** Recreation potential calculated with multicriteria evolution. Recreation potential among high capacity (category 1: green polygon), middle capacity (category 2: yellow polygon), and no capacity (red polygon).

*Landscape Approach*

Our results show that landscape attributes relevant for determining the recreation potential differ across regions of Varamin. Each of the landscape indicators explained in Table 4 was calculated in each of the three regions, and then the final map resulting from the combination of these indicators was prepared for each region (Figure 7).

In Figure 7, the potential of outdoor recreation in each of the three parts of the studied area is shown based on a number (0 as the lowest to 1 as the highest). This number was obtained from the combination of the six indicators of the landscape explained in Table 4. The output results of the landscape method were somewhat similar to the multicriteria evaluation method, such that, similar to the output of the multicriteria evaluation method, in the landscape method, many parts in the Shokrabad region are known to have recreation potential, except parts of the north and northwest.

Tables 6 and 7 show the areas of each of the different classes, and the suitability of the studied areas for extensive recreation for both multivariate evaluation methods and landscape approach, respectively. Both methods obtained almost similar results, but there are also differences. For example, in the case of the Fakhrabad region, in the multicriteria evaluation method, there is no area in the high suitability class, while with the landscape method, about 0.28 km² of the region is in this class. In general, it seems that with the landscape method, a higher area of the three regions are placed in the classes of high and middle capacity.

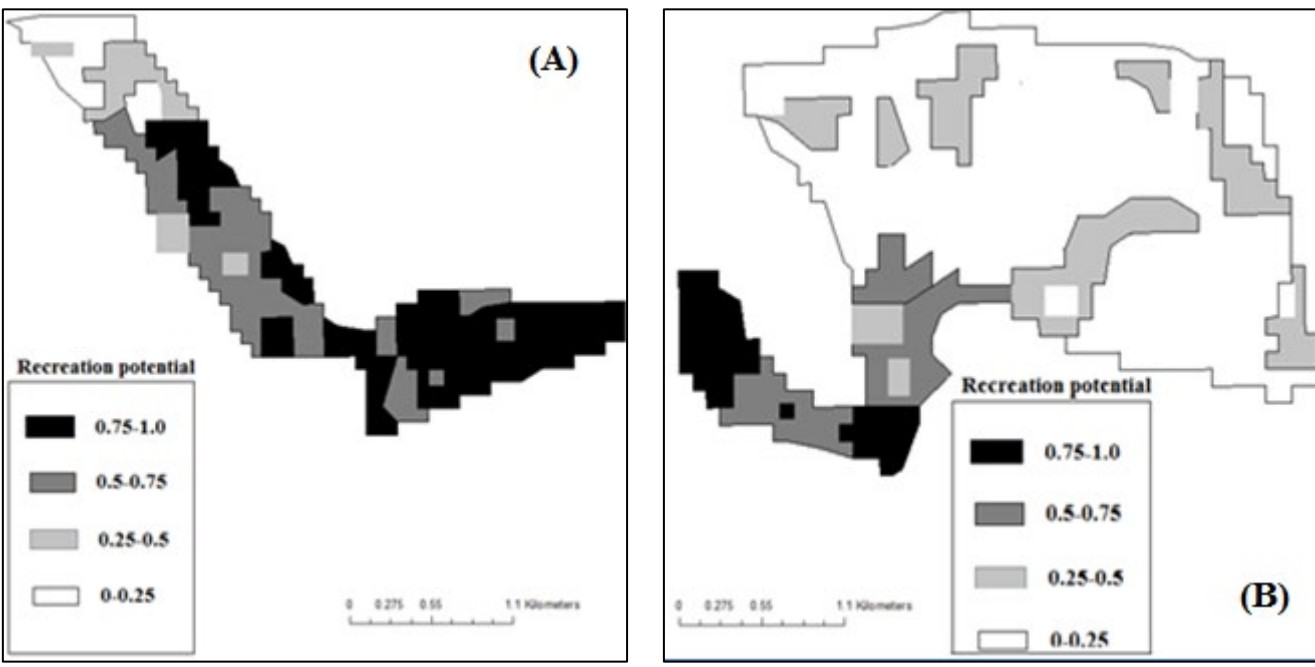

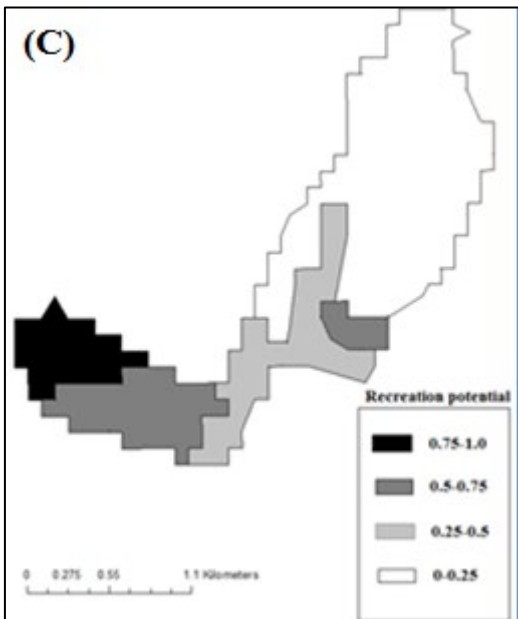

**Figure 7.** Recreation potential calculated with landscape indicators: (**A**) Shokrabad region; (**B**) Dolatabad region; (**C**) Fakhrabad region. Recreation potential is expressed on a 0–1 scale, where 0 is no recreation potential and 1 is maximum recreation potential.

**Table 6.** The area (km$^2$) of different classes of recreation potential (with multicriteria evolution).

| Region | High Capacity | Middle Capacity | No Capacity |
| --- | --- | --- | --- |
| Shokrabad | 2.11 | 1.49 | 0.51 |
| Fakhrabad | 0 | 0.37 | 2.66 |
| Dolatabad | 0.05 | 0.44 | 5.78 |

**Table 7.** The area (km$^2$) of different classes of recreation potential (with landscape approach).

| Region | Class 1 (0.75–1) | Class 2 (0.75–0.5) | Class 3 (0.5–0.25) | Class 4 (0–0.25) |
|---|---|---|---|---|
| Shokrabad | 1.87 | 1.62 | 0.18 | 0.44 |
| Fakhrabad | 0.28 | 0.53 | 0.50 | 1.72 |
| Dolatabad | 0.30 | 0.1 | 0.75 | 5.12 |

## 4. Discussion

In the present study, GIS was used for mapping the ecological factors and integrating these maps. None of the three studied areas, namely, Shokrabad, Fakhrabad, and Dolatabad, had any special restrictions in terms of topographic factors, i.e., slope, aspect, and elevation, as the elevation range in all three areas was between 700 to 850 m. Additionally, in terms of slope, these areas are among the very low slope areas, as the region as a whole varied between 1 to 5 percent, and this slope will not create restrictions for most recreational conditions and development. Therefore, these two factors of topography are not limiting factors. Geographically, they are slightly more diverse than the other two topographic factors. In the region there are all four main aspects, namely, southeast and west, and the aspect factor may be considered the only topographic limiting factor for tourism development. The geographical aspect factor is one of the important ecological factors in the suitability and capability of an area for recreation (wide and concentrated). These are known as the first floor for the wide recreation of the east and north aspects, and as the potential of the second floor for the west and south aspects [42,53,54].

The results of this research showed that only small parts of the Shokrabad region and even less of the Dolatabad region have the inherent ability to support outdoor recreation. Among the investigated factors, since the factors of elevation, slope, and aspect, as well as climate, had little variability in almost all three regions, the determining factors of recreation suitability in this research were soil and vegetation. In terms of the type of vegetation suitable for outdoor recreation, and the density of vegetation, all three studied areas had limitations and weaknesses, but to some extent, the Shokrabad region had a better situation in this regard; both the type and the density of the vegetation in it, compared to the two regions, were a little more appropriate. Additionally, the results of laboratory analyses of soil samples confirmed this and it was found that in terms of overall fertility and soil nutrients, the Shokrabad region has better suitability than the other two regions. In addition, in terms of structure and soil texture, which is an important factor in examining the potential of a region for tourism [55], the Shokrabad region has potential in this field, but the Fakhrabad and Dolatabad regions do not have much potential in this field. According to the outdoor recreation model, the optimal limit for the first floor is between 40 to 80% of the vegetation density. In terms of vegetation, only a part of the Shokrabad area and a small part of the other two areas had recreational capacity, and other parts did not have the necessary utility in this regard (Figure 5). Another important factor that was mapped and studied was the climatic factor, namely, precipitation and temperature in the region, which, due to the small size of the study area, was not a very restrictive factor and did not have significant changes in the entire study area. Thus, the amount of precipitation in the three regions varied between 20 and 100 mm, which, in this regard, according to meteorological station statistics, only the Shokrabad region had slightly higher precipitation and relative humidity than the other two regions, especially the Fakhrabad region. In terms of average annual temperature, the temperature varies between 20 to 25 degrees, and for the two climatic factors studied, namely, temperature and precipitation, the situation in the Shokrabad region is slightly better than the other two regions. In general, in terms of annual precipitation, favorable conditions are not provided. Thus, a complete and comprehensive study of the soil of the region was performed; the soil texture varies between loamy clay and silty loamy clay in different parts of the three regions, and parts with loamy soil and loamy clay used for development and leisure have no restrictions. Other soil factors analyzed, including nutrients and organic matter content, are not important for recreation and are

important for other assessments, especially species planting and afforestation. Additionally, according to the study of the land use map of the region and other basic maps of the region, since there was no significant fault near or in close proximity to the study area, from the perspective of this factor, there was no limitation and it was not included in the evaluation.

The results also showed that the amount of area with extensive recreational potential in the landscape approach in comparison to the multicriteria evaluation method was greater, and this method was easier to implement. In the landscape approach, we combined the six features of the landscape that we fully explained in Table 4. Recently, the landscape approach has been used in many environmental studies, including the estimation of the tourism potential as suggested by these studies [18,56,57]. The average annual temperature was a factor that has a significant effect on the touristic value of an area. A high level of this factor indicates areas with harsh weather conditions, and harsh weather is an important factor that campers pay much attention to in their extensive recreation [49]. For this reason, in some parts of the study area (especially the Dolatabad region), recreation potential may be underestimated (Figure 6). Similarly, Weyland and Laterra [18] illustrated similar results, suggesting that some parts of Argentina are unsuitable for recreation due to harsh weather conditions. Rivers, lakes, and ponds attract recreationists, and for this reason, this is one of the factors in determining the recreation potential of an area [15,52]. Although, in general, the studied area is not very rich in terms of the presence of permanent rivers, in some parts of Shokrabad, the existence of rivers and ponds, on the one hand, has improved the climate and soil conditions of the region, and, on the other hand, has increased the interest of tourists in this area. In terms of this factor, this region has a much better situation than the other two regions. Another important factor that was examined in the landscape approach was the state of density of tree cover and the amount of bare land. As a result, according to tourists, the entertainment value of the area increases. The highest percentage of tree cover in the study area was in the Shokrabad region, especially in the south and east of the region (the density of tree cover is about 40% and higher), and parts of the Fakhrabad region also had a medium percentage of tree cover (between 20 and 40%). In this respect, Dolatabad was poorer than the other two regions. In general, by using the landscape approach, it was found that parts of the Shokrabad region have high recreational potential (Figure 6), whereas only a smaller number of places in the Fakhrabad and Dolatabad regions show potential. However, in general, the multicriteria evaluation (Figure 5) yielded a higher amount of areas with recreational potential than the output of the second approach, i.e., the landscape method (Figure 6), which could be because more factors were considered and examined in the multicriteria evaluation method, and topography and soil factors also played a role in the evaluation.

Each of these methods of evaluating the recreation potential that were examined in this research has its advantages and disadvantages, which were also illustrated in other studies, and may vary depending on the purpose of the study and the type of region [28,58–60]. However, they may be used in a complementary fashion. Additionally, each region may require a specific method and specific indicators according to its own characteristics; for example, in forest areas, the cover factor is likely important.

Finally, an ecological evaluation of three regions of the Gaz forest of Varamin city (Shokrabad, Fakhrabad, and Dolatabad) was carried out. In the Shokrabad region, the southern, western, and eastern parts show a higher capacity for outdoor recreation and the northern part is more limited, probably due to limited ecological factors inclusion such as aspect, soil, vegetation, etc. The Fakhrabad region, due to unfavorable ecological conditions (poor soil with very high electrical conductivity, very low nutrients, low rainfall, and lack of vegetation), in general, did not have the necessary conditions for recreational development, and this situation was less severe in the Dolatabad region. The Shokrabad region has more potential than the other two regions (Fakhrabad and Dolatabad) in terms of the sum and ecological factors affecting the tourism potential. However, in terms of topographic factors, i.e., slope, aspect, and elevation, the three regions were almost similar. This is because the

general appearance of the region is relatively flat and the variability of topographic factors and climatic factors is relatively low.

## 5. Conclusions

Utilizing the capabilities of tourism and ecotourism in any region can provide society a dynamic and active ground for the development of the region (Bricker and Kerstetter, 2005). Therefore, there is a special need for the evaluation of ecotourism potential and the analysis of these capabilities in a geographical manner. For this purpose, in this study, potential recreation site assessment was performed by combining the most important ecological factors including elevation, slope, aspect, vegetation status, temperature, precipitation, and soil. Assessing the capabilities of a proposed site or area is a necessary and important issue for any type of development. The process of assessing a site's capabilities for development mainly consists of two steps: the first is to determine the most important limiting factors and the second is to evaluate the site (select the most appropriate parts based on limiting factors) using GIS, which has the ability to assist in assessing the potential of sites for tourism. Therefore, the questions at the beginning of this research can be answered in that in the entire studied area, only a few parts of Shokrabad, according to Figure 6, had the ability to develop outdoor recreation. Among the investigated ecological factors, since the variability of climatic and topographical factors was relatively low among the three regions, the soil factor determined the recreational potential. In the comparison of the two methods that were used in this research to evaluate recreational potential, as seen in the final results, the output of both methods was almost the same, with the difference being the amount of areas with extensive recreational potential. The landscape method indicated more potential recreation areas, and this method was easier to implement. As a result, both methods might be used as a supplement to evaluate the power of extensive recreation.

**Author Contributions:** M.B. conceived and designed the experiments; M.B. and S.H.M. performed the experiments and analyzed the data; S.H.M., M.B., P.B., H.U.M.A., S.K.H. and A.A. wrote the paper. All authors have read and agreed to the published version of the manuscript.

**Funding:** This research was funded by Iran National Science Foundation (INSF), grant number 4000690.

**Institutional Review Board Statement:** Not applicable.

**Informed Consent Statement:** Not applicable.

**Data Availability Statement:** Data will be made available on request.

**Acknowledgments:** The authors thank all persons who helped to collect data and acknowledge all funding institutions that supported the study through funding.

**Conflicts of Interest:** The authors declare that they have no known competing financial interests or personal relationships that could have appeared to influence the work reported in this paper.

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
