# Peer review of "Recreation Potential Assessment at Tamarix Forest Reserves: A Method Based on Multicriteria Evaluation Approach and Landscape Metrics"

_forests, doi:10.3390/f14040705_

Round 1
Reviewer 1 Report
In this paper, the authors develop new methods for describing outdoor recreation potential based on landscape indicators and systemic multi-criteria evolution in the context of Gaz forest of Varamin city. This is an interesting research paper with good design, but not fit for publication in its current form. My recommendation is to return the paper to the authors with an invitation to re-resubmit the paper after major revision.
My specific comments are as follows:
1) Line 38: Cite relevant source for this statement.
2) Line 45-47: Why “there is currently a demand for quantitative and computational methods based on geographical and quantitative data”? The authors need to provide a clear reasoning here.
3) Line 56: Not sure if “identify and acknowledge” is the suitable word here since both words convey similar meanings. Maybe “identify and integrate” would be more appropriate given the context of the sentence.
4) Line 70: The meaning of the sentence “Several studies have been conducted that assessed outdoor recreation approach around the world” is not clear. What does ”assessed outdoor recreation approach” mean? Do you mean “Several studies have been conducted to assess outdoor recreation potential around the world”?
5) Line 111: Add a comma “,” after “at present”.
6) Line 117-118: “Evaluating the suitability of an area for recreational activities requires a large amount of spatial data that includes various ecological, economic, and social factors.”. Besides, ecological, economic and social factors, there are also transport factors as it directly influences whether people can access the recreation. Also, consider adding some citations such as the ones below to support this statement.
a. Grunewald, K., Richter, B., Meinel, G., Herold, H. and Syrbe, R.U., 2017. Proposal of indicators regarding the provision and accessibility of green spaces for assessing the ecosystem service “recreation in the city” in Germany. International Journal of Biodiversity Science, Ecosystem Services & Management, 13(2), pp.26-39.
b. Gül, A., Örücü, M.K. and Karaca, Ö., 2006. An approach for recreation suitability analysis to recreation planning in Gölcük Nature Park. Environmental management, 37, pp.606-625.
c. Liu, D., Kwan, M.P. and Kan, Z., 2021. Analysis of urban green space accessibility and distribution inequity in the City of Chicago. Urban Forestry & Urban Greening, 59, p.127029.
7) Figure 1: This figure is hard to understand. First of all, does the red or the light blue part of the Figure 1(A) represent the Varamin City? There should be a legend in Figure 1(A) marking which color represents which city/county. Besides, for the three regions in Figure (B) (C) (D), I’d putting them into one single map and differentiate them by different colors (again with a legend).
8) Figure 2: The cases used in the figure are not uniform. For example: the last word in “Class Area”, “Core Area” and “Annual Temperature” are upper-case; while the last word in “Number of patches” and “Tree cover” is lower-case. The authors may want to change the case format to make it uniform.
9) Line 168: Consider creating a table for all the criteria (elevation, slope, direction, soil texture, vegetation density, soil fertility, climatic factor). On the one side, list the names of the criteria; on the other side, list a brief description of the criteria. This can improve the readability significantly.
10) Figure 7: Lack of scale bar and north arrow.
11) Figure 7: Again, the legend for different maps within this figure doesn’t appear to be uniform in font size. Moreover, parts of the second map of this figure is covered and not fully displayed.
12) Line 180: Year of the DEM data.
13) Line 287/Line 312/Line 317: Lack of dash line (“-”) between “multi” and “criteria”. The format (“multi-criteria”) should be uniform throughout the manuscript.
14) Line 297-298: “Each of the landscape indicators explained in Table 4 were calculated in each of the three regions…” should be “Each of the landscape indicators explained in Table 4 was calculated in each of the three regions…”
15) Line 323-324 of Page 14: “Geographic Information System (GIS) was used in mapping the 323 ecological factors and integrates these maps.” should be “Geographic Information System (GIS) was used in mapping the ecological factors and integrating these maps.”
16) Conclusions section: I don’t see any discussion about the limitation/future direction in the Conclusions section. The authors may want to consider adding these in the final section.
17) The format of the reference items is not uniform. For example:
a. Brabyn, L., & Mark, D.M. (2011). Using viewsheds, GIS, and a landscape classification to tag landscape photographs. Applied Geography, 31, 1115-1122. doi:10.1016/j.apgeog.2011.03.003
b. Brown, G., and Weber, D., 2011. Public participation GIS: A new method for national park planning. Landscape and Urban 474 Planning, 102(1), 1-15.
Why does reference item (a) has bracket for its publication year but item (b) does not? Why does item (a) has doi but item (b) does not? The authors should be careful with simple mistakes like this by conducting a careful proofreading throughout the entire revised manuscript before re-submission.
Author Response
Dear Dr. Editor
Thank you for considering a revised version of our previous paper "Recreation potential assessment at Tamarix forest reserves: A method based on multi-criteria evaluation approach and landscape metrics” that incorporates the many useful comments and suggestions of the reviewers. Detailed corrections are listed below point by point. Based on the reviewer comments, we have substantially modified the paper. Also, the English of the article was completely revised by Dr. Bettinger.
Yours sincerely,
Reviewer 1: In this paper, the authors develop new methods for describing outdoor recreation potential based on landscape indicators and systemic multi-criteria evolution in the context of Gaz forest of Varamin city. This is an interesting research paper with good design, but not fit for publication in its current form. My recommendation is to return the paper to the authors with an invitation to re-resubmit the paper after major revision.
Thank you very much for your warm comment.
Line 38: Cite relevant source for this statement.
- We revised the manuscript as you suggested:
In the past, the suitability of an area for extensive outdoor recreational activities was often evaluated with methods that required photo collection, surveys, and interviews, such as checklists or traditional methods of manual intersecting of ecological maps (Winter et al., 2019).
2) Line 45-47: Why “there is currently a demand for quantitative and computational methods based on geographical and quantitative data”? The authors need to provide a clear reasoning here.
- The reason for this sentence was added with the reference:
Despite the fact that these methods are useful for evaluating outdoor recreational activities, there is currently a demand for quantitative and computational methods based on geographical and quantitative data. Classic methods involving quantitative databases can be used to evaluate the process of landscape change and potential future conditions, and by using these databases, it may be possible to judge the outcomes of potential changes and the success or failure of management decisions. These methods are sometimes more accurate and precise than contemporary computer-based methods and may produce results closer to reality. However, both classic and new methods may be complementary and both may be needed to evaluate the recreation potential of a place (Shepard, 2005).
3) Line 56: Not sure if “identify and acknowledge” is the suitable word here since both words convey similar meanings. Maybe “identify and integrate” would be more appropriate given the context of the sentence.
We revised the manuscript as you suggested
4) Line 70: The meaning of the sentence “Several studies have been conducted that assessed outdoor recreation approach around the world” is not clear. What does ”assessed outdoor recreation approach” mean? Do you mean “Several studies have been conducted to assess outdoor recreation potential around the world”?
We revised the manuscript as you suggested
5) Line 111: Add a comma “,” after “at present”.
We revised the manuscript as you suggested
6) Line 117-118: “Evaluating the suitability of an area for recreational activities requires a large amount of spatial data that includes various ecological, economic, and social factors.”. Besides, ecological, economic and social factors, there are also transport factors as it directly influences whether people can access the recreation. Also, consider adding some citations such as the ones below to support this statement.
- Grunewald, K., Richter, B., Meinel, G., Herold, H. and Syrbe, R.U., 2017. Proposal of indicators regarding the provision and accessibility of green spaces for assessing the ecosystem service “recreation in the city” in Germany. International Journal of Biodiversity Science, Ecosystem Services & Management, 13(2), pp.26-39.
- Gül, A., Örücü, M.K. and Karaca, Ö., 2006. An approach for recreation suitability analysis to recreation planning in Gölcük Nature Park. Environmental management, 37, pp.606-625.
- Liu, D., Kwan, M.P. and Kan, Z., 2021. Analysis of urban green space accessibility and distribution inequity in the City of Chicago. Urban Forestry & Urban Greening, 59, p.127029.
We revised the manuscript as you suggested
Evaluating the suitability of an area for recreational activities requires a large amount of spatial data that includes various ecological, economic, and social factors (the accessibility of the area, the condition of the roads, the type and number of roads, and the general condition of transportation). (Gül, et al., 2006; Liu, et al., 2021).
7) Figure 1: This figure is hard to understand. First of all, does the red or the light blue part of the Figure 1(A) represent the Varamin City? There should be a legend in Figure 1(A) marking which color represents which city/county. Besides, for the three regions in Figure (B) (C) (D), I’d putting them into one single map and differentiate them by different colors (again with a legend).
- We revised the Figure 1 as you suggested:
8) Figure 2: The cases used in the figure are not uniform. For example: the last word in “Class Area”, “Core Area” and “Annual Temperature” are upper-case; while the last word in “Number of patches” and “Tree cover” is lower-case. The authors may want to change the case format to make it uniform.
- Figure 2 revised as suggested
9) Line 168: Consider creating a table for all the criteria (elevation, slope, direction, soil texture, vegetation density, soil fertility, climatic factor). On the one side, list the names of the criteria; on the other side, list a brief description of the criteria. This can improve the readability significantly.
- It was revised and according to the opinion of the reviewer, ecological factors were listed. Also, this part was shortened and the table 1 was added to this part and the information inside the table was brought.
Table 1. Ecological factors used to evaluating outdoor recreation potential
Variable |
Source |
Expected effect on recreation potential |
Elevation |
The DEM map (year of 2020) was extracted from the https://urs.earthdata.nasa.gov/ site |
Elevation is an important feature in environmental studies because it has a direct effect on other ecological factors such as precipitation and temperature (Bayat et al., 2019). |
Slope |
DEM, along with ArcGIS 10.3 software, to develop a slope map |
The morphology and slope of the basin is one of the main parameters in environmental studies that in addition to a direct effect on the intensity of water flow, also affects other physical characteristics of the basin (Bayat et al., 2021a). |
Aspect |
DEM and the Aspect tool was used in the ArcMap environment, and after creating the aspect map and necessary corrections, such as removing very small directions and merging them with larger directions |
. The aspect of the domain and its changes play an important role in changing its environment. The most important effect of the aspect of the slope manifests itself in the form of differences in weather conditions (Bayat et al., 2021b). |
Temperature and precipitation |
Climate map of Varamin meteorological stations and precipitation. The temperature and precipitation map was taken from the regional meteorological department included the isolines with ArcMap |
Climate is one of the most important geophysical factors that play an important role in the establishment or non-establishment of human settlements. Therefore, the climate is an important criterion in the establishment of tourist centers (Martín, 2005). |
Vegetation density |
Field visiting, report and mapping by ArcMap |
The higher the vegetation density, the higher the tourism potential for this purpose (Arbieu, et al., 2017)
|
Soil |
Field work, sampling with plots, lab analysis and mapping by ArcMap |
Soil erosion is one of the most fundamental environmental, agricultural, and food production problems in the world that has devastating effects on all natural and fabricated ecosystems. Soil loss is therefore recognized as a serious environmental problem (Cetin et al., 2018). |
10) Figure 7: Lack of scale bar and north arrow.
- We revised the Figure 7 as you suggested
11) Figure 7: Again, the legend for different maps within this figure doesn’t appear to be uniform in font size. Moreover, parts of the second map of this figure is covered and not fully displayed.
We revised the Figure 7 as you suggeste
12) Line 180: Year of the DEM data.
- revised as suggested: The DEM map (year of 2020) was extracted from the https://urs.earthdata.nasa.gov/ site with a resolution of 12.5 meters and entered into the ArcMap environment. This map was the base map of slope and aspect maps.
13) Line 287/Line 312/Line 317: Lack of dash line (“-”) between “multi” and “criteria”. The format (“multi-criteria”) should be uniform throughout the manuscript.
14) Line 297-298: “Each of the landscape indicators explained in Table 4 were calculated in each of the three regions…” should be “Each of the landscape indicators explained in Table 4 was calculated in each of the three regions…”
We revised the manuscript as you suggested
15) Line 323-324 of Page 14: “Geographic Information System (GIS) was used in mapping the 323 ecological factors and integrates these maps.” should be “Geographic Information System (GIS) was used in mapping the ecological factors and integrating these maps.”
We revised the manuscript as you suggested
These approaches involve a geographical analysis for selecting the most suitable places, and often this involves the use of Geographic Information System (GIS) and methods for overlapping ecological factors (Berry et al. 1991).
16) Conclusions section: I don’t see any discussion about the limitation/future direction in the Conclusions section. The authors may want to consider adding these in the final section.
The discussion and conclusion section was revised and the important results that were obtained in this research were added in a specific way in these sections.
17) The format of the reference items is not uniform. For example:
- Brabyn, L., & Mark, D.M. (2011). Using viewsheds, GIS, and a landscape classification to tag landscape photographs. Applied Geography, 31, 1115-1122. doi:10.1016/j.apgeog.2011.03.003
- Brown, G., and Weber, D., 2011. Public participation GIS: A new method for national park planning. Landscape and Urban 474 Planning, 102(1), 1-15.
Why does reference item (a) has bracket for its publication year but item (b) does not? Why does item (a) has doi but item (b) does not? The authors should be careful with simple mistakes like this by conducting a careful proofreading throughout the entire revised manuscript before re-submission.
We revised the references as you suggested and according MDPI format.

Reviewer 2 Report
1. Lines 164-260, the description of "Ecological factors affecting tourism development" is too detailed. I suggest that they be appropriately streamlined. At the same time, the structure arrangement is not very standardized, and it is recommended to modify it according to the journal format.
2. What grid or spatial unit is the manuscript based on for recreation potential evaluation and spatial mapping? What is the resolution of the ecological data? The manuscript needs to explain this. This allows reviewers to assess the feasibility of the data.
3. The resolution of the final evaluation results presented by the manuscript (Figure 7) is not fine, is the case area scale too small?
4. The discussion of the manuscript is fragmented, and I recommend summarizing it into 3-4 points rather than explaining each indicator.
Author Response
Reviewer 2:
- Lines 164-260, the description of "Ecological factors affecting tourism development" is too detailed. I suggest that they be appropriately streamlined. At the same time, the structure arrangement is not very standardized, and it is recommended to modify it according to the journal format.
- It was revised and according to the opinion of the reviewer, ecological factors were listed. Also, this part was shortened and the table 1 was added to this part and the information inside the table was brought.
- What grid or spatial unit is the manuscript based on for recreation potential evaluation and spatial mapping? What is the resolution of the ecological data? The manuscript needs to explain this. This allows reviewers to assess the feasibility of the data.
-These maps were prepared from the DEM image that had a spatial resolution of 12.5 meters, and therefore their resolution is 12.5 meters.
- The resolution of the final evaluation results presented by the manuscript (Figure 7) is not fine, is the case area scale too small?
We revised the manuscript as you suggested
- The discussion of the manuscript is fragmented, and I recommend summarizing it into 3-4 points rather than explaining each indicator.
We revised the manuscript as you suggested
- The discussion and conclusion section was revised and the important results that were obtained in this research were added in a specific way in these sections.

Reviewer 3 Report
The work submitted for review is an interesting scientific article with a correct structure. The language of the work is also generally correct, although some phrases are unclear.
Overall, I evaluate the work positively. However, in my opinion, some changes are necessary to improve the quality of the work.
1. in the introduction, I propose to clarify what methods the authors have in mind in the sentence: "In the past, the suitability of an area for extensive outdoor recreational activities was often evaluated with methods that required photo collection, surveys, and interviews." In addition, I suggest clarifying the term "ecosystem services approach" and "landscape metrics" with reference to relevant literature.
2. in the description of the study area, I propose to add characteristics of the analyzed forest reserves in terms of species composition, land use, etc.
3. Fig.1 should be improved by signing A, B, C, D; besides, I propose to add a map with more detailed location of the area. Fig.1 ABCD are of poor quality. It is not clear what is the relationship of Fig.A and Ryc.BCD. The last ones only show the shape of the analyzed areas and the altitude. This is a bit too little.
4. at the figures and tables, the origin of the information should be added
5.Fig.6. and Fig.7 should be signed, which region they refer to
6. I suggest that the sentence "In terms of vegetation type and density factor, these three areas were more limited, as in Shokrabad region, the situation was better in terms of this factor" (line 338-339) should be made more specific, as the wording used in it is unclear
7. the conclusions should address all the research questions (lines 131-140) and not just selected ones
Author Response
Reviewer 3:
- in the introduction, I propose to clarify what methods the authors have in mind in the sentence: "In the past, the suitability of an area for extensive outdoor recreational activities was often evaluated with methods that required photo collection, surveys, and interviews." In addition, I suggest clarifying the term "ecosystem services approach" and "landscape metrics" with reference to relevant literature.
We revised the manuscript as you suggested
The ecosystem approach makes explicit the link between the status of natural resource systems and ecosystem services that support human well-being (García-Llorente, et al., 2018) and landscape metrics are measurable units of landscape composition and act as a surrogate for change, thus allowing for the description and quantification of spatial pat-terns and ecological processes over time and space (Herold et al., 2002).
in the description of the study area, I propose to add characteristics of the analyzed forest reserves in terms of species composition, land use, etc.
-According to the opinion of the reviewer, the composition of species was added to this section:
In this area, 5 plant types were identified as follows:
Type 1: Artemisia sieberi + Salsola tomentosa + Tamarix spp
In this type (abbreviated as: Ar. si. + Sa. to. + Ta. Spp.), plain heather, salt grass, and sedge were identified as dominant species.
Type 2: Haloxylon persicum + Annual plants + Artemisia sieberi
In this type (abbreviation: Ha. pe. + Annual plants + Ar. Si) Zardtagh, all kinds of one-year species and plains sedge were identified as dominant species.
Type 3: Artemisia sieberi + Alhagi maurorum
In this abbreviated type (Ar. si. + Al. ma), the Iranian plain and Kharshter were identified as dominant species.
Type 4: Artemisia sieberi + Salsola tomentosa + Calligonum comosum + Seidlitzia rosmarinus
In this type (abbreviated as: .Ar. si. + Sa. to. + Ca. co. + Se. ro), plains grass, salt grass, skanbil and ashnan were identi-fied as dominant species.
Type 5: Salsola spp. + Aeluropus littoralis
In this abbreviated type (Sa. spp. + Ae. Li), types of grass and grass were identified as dominant species.
Fig.1 should be improved by signing A, B, C, D; besides, I propose to add a map with more detailed location of the area. Fig.1 ABCD are of poor quality. It is not clear what is the relationship of Fig.A and Ryc.BCD. The last ones only show the shape of the analyzed areas and the altitude. This is a bit too little.
We revised the Figure 1 as you suggested
- at the figures and tables, the origin of the information should be added
We revised the manuscript as you suggested
5.Fig.6. and Fig.7 should be signed, which region they refer to
We revised the manuscript as you suggested
- I suggest that the sentence "In terms of vegetation type and density factor, these three areas were more limited, as in Shokrabad region, the situation was better in terms of this factor" (line 338-339) should be made more specific, as the wording used in it is unclear
We revised the manuscript as you suggested
In terms of the type of vegetation suitable for outdoor recreation, and the density of vegetation, all three studied areas had limitations and weaknesses, but to some extent, the Shokarabad region was in a better situation in this regard. That is, both the type and the density of the vegetation in the Shokarabad region, compared to the other two regions, was a little more appropriate.
- the conclusions should address all the research questions (lines 131-140) and not just selected ones
- We revised the conclusion section and tried to answer all the questions at the beginning of the article in this section, according to the opinion of the reviewer:
Therefore, the questions at the beginning of this research can be answered in this way that in the entire studied area, only a few parts of Shokrabad according to the figure 6 had the ability to develop outdoor recreation. Among the investigated ecological factors, since the variability of climatic and topographical factors was relatively low among the three regions, the soil factor determined the recreational potential. In the comparison of the two methods of evaluating the recreational potential that were used in this research and the landscape method, as seen in the final results, the output of both methods was almost the same, with the difference that the amount of areas with extensive recreational potential. The landscape method indicated more potential recreation areas, and, this method was easier to implement. As a result, both methods can be used as a supplement to evaluate the power of extensive recreation.

Round 2
Reviewer 1 Report
I appreciate the authors’ efforts in addressing my comments. The revised manuscript has been improved significantly. I am now satisfied with authors’ revision and have no further comments. Therefore, I would recommend accepting the paper for publication.
Author Response
Dear Dr. Editor
Thank you for considering a revised version of our previous paper "Recreation potential assessment at Tamarix forest reserves: A method based on multi-criteria evaluation approach and landscape metrics”
I checked the references and changed the references that are not relevant.
Konijnendijk, C.C. Evidence-based guidelines for greener, healthier, more resilient neighbourhoods: Introducing the 3–30–300 rule. Journal of Forestry Research 2022, 1-10.
I changed this reference with the following reference:
Herzberg, R.; Pham, T.G.; Kappas, M.; Wyss, D.; Tran, C.T.M. Multi-Criteria Decision Analysis for the Land Evaluation of Potential Agricultural Land Use Types in a Hilly Area of Central Vietnam. Land 2019, 8, 90. https://doi.org/10.3390/land8060090
Masteali, S.H., Bettinger, P., Bayat, M., Amiri, B.J. and Awan, H.U.M.,. Comparison between graph theory connectivity indices and landscape connectivity metrics for modeling river water quality in the southern Caspian sea basin. Journal of Environmental Management, 2023, 328, p.116965.11.
I changed this reference with the following reference:
Kandari, A.M., Kasim, S., Limi, M.A. and Karim, J.,. Land suitability evaluation for plantation forest development based on multi-criteria approach. Journal Agriculture, Forestry and Fisheries., 2015, 4(5), pp.228-238.
Bayat, M., Namiranian, M. and Zobeiri, M., Volume, height and wood production modeling using the changes in a nine years rotation (case study: Gorazbon district in Kheyroud forest, north of Iran). Forest and Wood Products, 2014, 67(3), pp.423-435.
I changed this reference with the following reference:
Wajchman-Świtalska, S., Zajadacz, A., Woźniak, M., Jaszczak, R. and Beker, C.,. Recreational Evaluation of Forests in Urban Environments: Methodological and Practical Aspects. Sustainability, 2022., 14(22), p.15177.
